# Epigenetics: A Potential Mechanism Involved in the Pathogenesis of Various Adverse Consequences of Obstructive Sleep Apnea

**DOI:** 10.3390/ijms20122937

**Published:** 2019-06-15

**Authors:** Yung-Che Chen, Po-Yuan Hsu, Chang-Chun Hsiao, Meng-Chih Lin

**Affiliations:** 1Division of Pulmonary and Critical Care Medicine, Department of Medicine, Kaohsiung Chang Gung Memorial Hospital and Chang Gung University College of Medicine, Kaohsiung 83301, Taiwan; yungchechen@yahoo.com.tw (Y.-C.C.); hsupowan@yahoo.com.tw (P.-Y.H.); 2Department of Medicine, College of Medicine, Chang Gung University, Taouyan 33302, Taiwan; 3Department of medical research, Kaohsiung Chang Gung Memorial Hospital and Chang Gung University College of Medicine, Kaohsiung 83301, Taiwan; 4Graduate Institute of Clinical Medical Sciences, College of Medicine, Chang Gung University, Taouyan 33302, Taiwan

**Keywords:** obstructive sleep apnea, intermittent hypoxia with re-oxygenation, epigenetics, DNA methylation, histone modification, non-coding RNA

## Abstract

Epigenetics is defined as the heritable phenotypic changes which do not involve alterations in the DNA sequence, including histone modifications, non-coding RNAs, and DNA methylation. Recently, much attention has been paid to the role of hypoxia-mediated epigenetic regulation in cancer, pulmonary hypertension, adaptation to high altitude, and cardiorenal disease. In contrast to sustained hypoxia, chronic intermittent hypoxia with re-oxygenation (IHR) plays a major role in the pathogenesis of various adverse consequences of obstructive sleep apnea (OSA), resembling ischemia re-perfusion injury. Nevertheless, the role of epigenetics in the pathogenesis of OSA is currently underexplored. This review proposes that epigenetic processes are involved in the development of various adverse consequences of OSA by influencing adaptive potential and phenotypic variability under conditions of chronic IHR. Improved understanding of the interaction between genetic and environmental factors through epigenetic regulations holds great value to give deeper insight into the mechanisms underlying IHR-related low-grade inflammation, oxidative stress, and sympathetic hyperactivity, and clarify their implications for biomedical research.

## 1. Introduction

Affecting approximately 25% of men and 13% of women worldwide, obstructive sleep apnea (OSA) is characterized by recurrent episodes of upper airway collapse, which result in recurring arousal and de-saturation during sleep, which lead to sleep fragmentation and chronic intermittent hypoxia with re-oxygenation (IHR) injury. Un-treated OSA is associated with adverse consequences including hypertension, ischemic heart disease, diabetes mellitus, dementia, and depression, as well as poor outcomes of malignant neoplasms [1]. It is now well established that OSA promotes endothelial dysfunction, atherosclerosis, and insulin resistance, and is associated with an increased risk of coronary heart disease, systemic/pulmonary hypertension, diabetes mellitus, cognitive dysfunction, stroke, and motor/vehicle accidents [2]. Common symptoms of OSA include excessive daytime sleepiness, dozing non-refreshing sleep, choking episodes during sleep, nocturia, morning headache, poor attention, memory impairment, confusional parasomnia, and insomnia. Many people with OSA use continuous positive airway pressure (CPAP) as the first line treatment, but adherence rates remain unacceptably low, especially for those with insomnia or minimal symptoms [3]. CPAP clearly improves vigilance and cognitive function, reduces insulin resistance, and is the most effective in lowering blood pressure in OSA patients with refractory hypertension. But the use of CPAP was not associated with reduced risks of cardiovascular outcomes, diabetes mellitus, or death for patients with OSA in recent randomized control trials [4,5,6,7,8]. Thus, it is essential to develop novel pharmacological agents to counteract pathophysiolocal mechanisms responsible for OSA-related adverse consequences, namely oxidative stress, sympathetic activation and low-grade inflammation [4].

### 1.1. Epigenetic Mechanisms Underlying Sustained Hypoxia

Epigenetic mechanisms that affect genes include histone modifications, non-coding RNAs, and DNA methylation. These epigenetic regulations alter both the accessibility of genes for transcription factor binding and the rates of gene transcription. Epigenetic alterations have been associated with hypoxia in cancer and a highly complex hypoxia-epigenetic interaction is observed during carcinogenesis and tumor progression [9]. Epigenetic regulation of gene expression have been shown to contribute to the development of hypoxic pulmonary hypertension and its phenotypic variability [10]. The epigenome is proposed to harbor important clues regarding the molecular mechanisms underlying human adaptation or maladaptation to hypoxic environment at high altitude as well as to heart and kidney disease [11,12].

### 1.2. Epigenetic Mechanisms Underlying IHR Injury in OSA

In contrast to sustained hypoxia, IHR is associated with repetitive episodes of re-oxygenation injury, which is like ischemia re-perfusion injury. Cycling hypoxia leads to more severe cytotoxicity than chronic hypoxia through enhancing the production of reactive oxygen species (ROS) and the activation of several transcription factors, such as hypoxia inducible factor-1 (HIF-1) and nuclear factor kappa B (NF-κB). [13]. The peripheral blood immune cells, endothelium, and other end organ tissues adapts to oxidative stress from IHR by modulating their gene expression in OSA patients. In the past two decades, genetic association studies on OSA have identified only a few genetic polymorphisms related to the apnea hypopnea index (AHI) independently, such as TNF-α-308G/A and serotonin 2A receptor -1438G/A, even though about 30% to 40% of the variability in OSA severity can be explained by familial factors [14,15,16,17].

Emerging evidence suggests that epigenetic changes, defined as the heritable phenotype changes that do not involve alterations in the DNA sequence, are relevant to the development of OSA and its pathogenesis. In response to external stimuli, such as hypoxia, toxin, or infection, DNA methylation, and post-translational histone modifications render gene promoter or enhancer regions more or less permissive to interaction with the transcriptional machinery, while a plethora of non-coding RNAs, such as microRNAs (miRNAs) and long non-coding RNAs (lncRNAs), either transcriptionally silence or degrade targeted messenger RNAs. These epigenetic modifications form an integrated and highly complex regulatory network to control gene expressions. The reversibility of epigenetic modifications and the relative ease with which non-coding RNAs can be manipulated and augurs well the development of much-needed novel pharmaceuticals for treatment of OSA. In this brief review, we consider recent advances in our understanding of epigenetic regulations induced by IHR in the hope of gaining valuable insights concerning mechanisms of pathogenesis in OSA patients with or without various adverse consequences.

## 2. ROS-HIF-1-Endothelin and ROS-TLR-NF-κB Signaling Pathways in OSA

### 2.1. IHR Induces Systemic Inflammation and Sympathetic Hyperactivity through Enhancing ROS Production

IHR is a major characteristic of OSA, and associated with lots of biological processes, including increased cell apoptosis, excessive oxidant stress and overt inflammation. OSA is linked with increased oxidative stress characterized by increased superoxide anion release from circulating leukocytes due to up-regulation of nicotinamide adenine dinucleotide phosphate (NADPH) oxidase (NOX) enzymes and down-regulation of antioxidants enzymes, such as superoxide dismutase (SOD), catalase, and thioredoxin, leading to reduced nitric oxide bioavailability and increased lipid peroxidation [18,19]. Through the imbalance between (NOX)-dependent generation of ROS and (SOD)-dependent eradication of ROS, IHR enhances the stabilization and activity of the HIF-1α transcription factor, which is well known to promote adaptive and maladaptive responses to hypoxia, while IHR triggers the degradation of HIF-2α [20]. As a result, the imbalance between HIF-1α-dependent prooxidant and HIF-2α-dependent antioxidant enzymes leads to further increases in ROS. Neurotransmitters of O2-sensing carotid body sensory nerves and projected brainstem neurons also trigger the imbalance between HIF-1α and HIF-2α activity as well as between NOX and SOD activity, leading to increased ROS generation and sympathetic hyperactivity [21,22]. Prolonged IHR exposure results in increased pro-inflammatory M1 macrophage, type 1 T helper cell reaction, and neutrophil infiltration with enhanced release of ROS, while disruption of endothelial tight junctions leads to increased leukocyte leakage from vessels into surrounding tissues, causing inflammation and cell injury to various end organs [23,24,25,26,27,28,29]. In our recent study, we found that over-expression of pro-inflammatory formyl peptide receptor 1 and insufficiency of anti-inflammatory formyl peptide receptor 2 in association with defective lipoxin A4 and resolving D1 production were associated with disease severity of OSA and its adverse consequences [30].

### 2.2. ROS-HIF-1-Endothelin Signaling Pathway in OSA

HIF-1 is composed of the cytosolic HIF-1α and nuclear HIF-1β subunits, the former being O2-sensitive and translocating into nucleus to heterodimerize with the latter and form the functional HIF-1 transcription factor in response to hypoxia. HIF-1 can activate transcription of many down-stream genes, either adaptive or maladaptive. IHR trigger increased accumulation of ROS, which in turn activates mammalian target of rapamycin-dependent synthesis of HIF-1α and inhibits prolyl hydroxylases-dependent HIF-1α degradation. Among the various HIF-1 target gene products for adaptive responses to IHR, endothelin (ET-1) has vasoconstrictive, growth promoting, and pro-inflammatory properties, and is increased both in OSA patients with hypertension and animals exposed to IHR. ET-1 is also involved in several pathophysiological pathways associated with adverse consequences of OSA, including hyperglycemia, myocardial infarction, heart failure and stroke [31,32].

### 2.3. ROS-TLR-NF-κB Signaling Pathway in OSA

On the other hand, IHR can accelerate growth and vulnerability of atherosclerotic plaques, and induce lung and renal injury, which probably act by triggering the activation of toll-like receptor 2/4/6-NF-κB pro-inflammatory signaling and the release of increased C-reactive protein, interleukin (IL)-6, tumor necrosis factor (TNF)-α, and interferon-γ [33,34,35,36]. Substantial inflammatory cytokines and peroxidation lead to necrosis and apoptosis of end organ cells, which eventually results in gradual neurocognitive, metabolic, endothelial, and cardiovascular dysfunctions of OSA patients [19,28,37].

## 3. Histone Modifications

The nucleosome core is formed of two H2A-H2B dimers and a H3-H4 tetramer, with the latter protruding long tails that can be covalently modified with acetylation (ac) or methylation (me). Generally speaking, highly acetylated histones form more accessible euchromatin and tend to associate with active transcription. Deacetylated histones, in contrast, form less accessible heterochromatin and contribute to gene silencing. Histone acetylation and deacetylation reactions are typically catalyzed by enzymes with histone acetyl-transferase (HAT) and histone deacetylase (HDAC) activity, respectively. Histones can be methylated on lysine (K) or arginine residue only, but methylation is most commonly observed on K residue of the histone tails H3 and H4. Histone methylation can be associated with either transcriptional repression or activation, depending on the position of residues modified and the number of methyl groups. For instance, methylation of lysine 4 in H3 (H3K4me2/3), H3K36me2/3, and H3K79me2/3 are related to transcription activation, whereas H3K9 and H3K27 dimethylation/trimethylation (H3K9me2/3 and H3K27me2/3) are related to transcription suppression [38]. Histone methylation is mediated by chromatin remodelers that include histone methyltransferases, histone lysine demethylases (KDMs), and other histone-modifying enzymes.

### 3.1. Histone Modifications in Sustained Hypoxia

Up-regulation of the oxygen-dependent KDMs in sustained hypoxia is proposed to increase demethylation of methylated lysine residues, including global and HIF-1α-specific H3K9me3, H3K27me3, H3K4me3, and H3K36me3. On the other hand, sustained hypoxia enhanced the demethylation of H3K9me2 and H3K9me1, leading to provision of un-methylated H3K9 residues that are substrates for HATs and increased H3K9/H3K23/H3K14 acetylation [39,40,41,42,43,44]. Additionally, ischemia-dependent decrease of H4K16Ac affects gene expression in a cell-specific manner [45]. Furthermore, HDAC3-dependent deacetylation has been demonstrated to alter functional equilibrium in response to acute hypoxia stimuli [46,47].

H3K4me3 occurring in the HIF-1 transcriptional complex due to loss of KDM4A is involved in HIF-1α nuclear transport, while H3K9me3 accumulating at the HIF-1α locus leads to a decrease in HIF-1α mRNA [48]. A recent genome wide analysis in human umbilical vein endothelial cells by chromatin immunoprecipitation sequencing has shown a massive signal covering much of the ET-1 encoding gene locus for both H3K4me3 and H3K27ac [48,49,50]. Previous studies indicate a critical role for H4K16Ac, H3K4me3, H3K27ac, H3K9me3, and H3K9ac over the promoter regions in epigenetic activation of the NOX 1,2,4, and 5 genes for cellular senescence, endothelial dysfunction, and pulmonary hypertension [51,52]. HDACs, including HDAC1-6, have been shown to increase HIF-1α protein stability to promote HIF-1 transactivation. Several HIF-1α-dependent KDMs, including KDM2B, 3A, 4A/B/C, 5B, and 6B, have been identified [53]. HDAC4 over-expression led to increased apoptosis and death of cardiomyocytes under hypoxia stimuli, while HDAC4 sumoylation and degradation protect cardiomyocytes from hypoxia-induced apoptosis [46,54].

### 3.2. Abnormal Histone Modifying Enzyme Expressions in OSA Patients or in Response to IHR Experiments In Vitro 

Sirtuin 1 (SIRT1), a class III HDAC, regulates endothelial nitric oxide synthase (eNOS), is disrupted by oxidative stress and systemic inflammation, and involved in different aspects of cardiovascular disease, aging and stress resistance. OSA patients had lower SIRT1 protein levels in blood, which could be reversed with either 3-month nasal CPAP or mandibular advancement device treatment in parallel with the change in serum nitrate level or leukocyte telomere length [55,56]. In contrast, up-regulation of HDAC2 was reported to play a critical role in the mechanisms and consequences of OSA-induced perturbations in visceral fat tissue depots, while HDAC2 is required for interferon-stimulated gene expression and monocyte chemoattractant protein 2 secretion [57]. These results suggest that IHR-induced HDAC2 over-expression and SIRT1 under-expression may be associated with cardiovascular dysfunction in OSA (Table 1).

### 3.3. Histone Modification Patterns in Animals Exposed to CIH 

Chronic intermittent hypoxia (CIH) during the sleep period has been used as a useful animal model of OSA, and is defined as various cycles of IHR events per day for more than four weeks in most research. Macrophages isolated from aortas of mice exposed to CIH showed over-representation of the active histone mark (H3K9ac) in pro-inflammatory and oxidative stress signaling pathways, such as HIF-1, p53, NF-kB, tumor growth factor-β, forkhead box protein O4, and IL-6, while genes associated with over-representation of the repressive histone mark (H3K27me3) were related to anti-inflammatory and glutathione redox pathways which are protective against atherosclerosis, such as peroxisome proliferator-activated receptor/retinoid X receptor, and liver X receptor/retinoid X receptor activation [58]. These epigenetic changes occurred in parallel with the persisted recruitment of CD36+ high M1 macrophages to the aortic wall and the triggering of atherogenesis. These results indicate that histone modification-mediated activation of the oxidative stress and inflammatory pathways may be involved in the establishment of the IHR-induced endothelial dysfunction, atherosclerosis, and aortic remodeling in OSA (Table 1).

## 4. Non-Coding RNAs

A plethora of non-coding RNAs, such as miRNAs and lncRNAs, either transcriptionally silence or degrade targeted messenger RNAs. miRNAs are a class of small RNAs, which are duplexes ~22 nucleotides in length, and regulate messenger RNA stability and translation through initiation block/post-initiation block directly, or deadenylation /degradation indirectly. Pairing of miRNAs with complementary sequences of target mRNAs at 3′ un-translated region inhibits the gene translation. The human genome may encode around 600 miRNAs, which appear to target about 60% of the genes of humans, while each miRNA has, on average, roughly 400 conserved targets. LncRNAs are defined as RNA transcripts with lengths exceeding 200 nucleotides that are not translated into protein, and characterized by higher tissue and developmental stage specificity. More than 1800 human lncRNAs have been functionally annotated with experimental evidences showing the vital roles in regulating genome at multiple levels, including transcriptional activation or repression, post-transcriptional regulation, and chromatin modification [59].

### 4.1. Differentially Expressed miRNAs in OSA Patients 

In a pilot study, a singular pre-CPAP treatment cluster of three plasma miRNAs (down-regulations of miR-378a-3p and miR-100-5p, up-regulation of miR-486-5p) could predict blood pressure responses to CPAP treatment in patients with resistant hypertension and OSA [60]. MiR-664a-3p was shown to be down-regulated in OSA patients, and negatively correlated with both AHI and maximum carotid intima-media thickness, suggesting a potential of using circulating miR-664a-3p as a noninvasive marker of atherosclerosis in OSA [61]. MiR-130a was associated with the presence and progression of pulmonary hypertension in OSA patients, probably through down-regulating *growth arrest-specific homeobox* (*GAX*) gene, which has been shown to maintain contractile phenotype and monitors proliferation and migration of vascular smooth muscle cells [62]. A small-scale sequencing analysis revealed that expression levels of miR-485-5p, 107, and 199-3p were down-regulated in OSA patients, while miR-574-5p expression showed up-regulation, indicating a close relationship of the differentially expressed miRNAs with OSA [63]. Exosomal miRNA-630 was reported to be a putative key mediator of endothelial function and cardiovascular disease risk in children with OSA, probably through targeting nuclear factor erythroid-2-related factor (*Nrf2*), AMP kinase, and tight junction pathways [27]. These results indicate that the presence of endothelial dysfunction, atherosclerosis, and hypertension in OSA may be associated with up-regulations of miR-130a and miR-574, and down-regulations of miR-664a, miR-485, miR-107, miR-199, and miR-630 (Table 2).

### 4.2. Non-Coding RNA Changes in Animals Exposed to CIH or in Response to IHR In Vitro 

Recent studies found that several miRNAs could influence IHR process and affect hypoxia-induced cell apoptosis in end organs [64]. Many miRNAs up-regulated by hypoxia are direct targets of HIF-1α, HIF-2α, NF-κB, or their responsive genes, and possess a positive feedback loop to stabilize HIF-1α protein, while other miRNAs down-regulated by hypoxia commonly suppress the expression of HIF or inflammatory signaling and engage in protective mechanisms against IHR injury. For instance, mir-218 could contribute to IHR-induced apoptosis in aortic endothelial cells through Roundabout Guidance Receptor 1 (*ROBO1*)/*HIF-1α* pathway, while miR-365 could inhibit IHR-induced inflammation through targeting *IL-6* [64,65,66,67]. Up-regulated miR-26b and down-regulated miR-207 were shown to contribute to the CIH-induced cognitive impairments by the induction of pro-apoptosis B-cell lymphoma 2 (*Bcl2*) Associated X (BAX) and inactivation of anti-apoptosis *Bcl2*-related mitochondrial signaling pathway in hippocampus in a rat model [66]. MiR-223 was reported to have anti-angiogenic effect and attenuate pulmonary hypertension in a rat CIH model [62,68]. CIH treatment could induce atrial remodeling and fibrosis via miR-21/sprouty RTK signaling antagonist 1(*Spry1*)/extracellular regulated protein kinases (*ERK*)/matrix metallopeptidase 9 (*MMP-9*) and miR-21/phosphatase and tensin homolog (*PTEN*)/phosphoinositide 3-kinases (*PI3K*)/protein kinase B (*AKT*) pathways as demonstrated in two rat models, while miR-145/SMAD family member 3 (*Smad3*) signaling pathway might attenuate aortic remodeling and sympathetic nerve sprouting [69,70]. MiR-31 could promote cardiac hypertrophy through targeting protein kinase C epsilon (*PKC*ε), which has been shown to blunt cardiac pathophysiologic responses in response to chronic hypoxia by a HIF-1a-mediated mechanism [71]. MiR-155 was shown to promote oxidation and enhance the CIH-induced NLR family pyrin domain containing 3 (NLRP3) inflammasome pathway in renal tubular cells by inhibiting the targeted forkhead box protein O3 (*FOXO3a*) gene in a murine model [72]. A recent in vitro study found that the expressions of resistin (*RETN*), *TNF-α*, and C-C motif chemokine ligand 2 (*CCL2*) of mouse and human pre-adipocytes/adipocytes, were increased by IHR via down-regulation of miR-452, which may play a protective role in the development of insulin resistance [73]. Additionally, IHR could up-regulate the levels of selenoprotein P (*SELENOP*) and hepatocarcinoma-intestine-pancreas/pancreatitis-associated protein (*HIP*/*PAP*) in human hepatocytes to accelerate insulin resistance and promote cell proliferation, respectively, via a miR-203 mediated mechanism [74]. These results indicate that up-regulations of miR-218, miR-26b, miR-21, miR-31, and miR-155, and down-regulations of miR-365, miR-207, miR-223, miR-145, miR-203, and miR-452 may contribute to the development of CIH or IHR-induced cardiovascular remodeling, cognitive dysfunction, insulin resistance, and chronic kidney injury (Table 2).

A small-scale microarray study on the heart samples of rats exposed to eight weeks of CIH first identified 289 differentially expressed lncRNAs, among which three up-regulated lncRNAs (XR_596701, XR_344474, and ENSRNOT00000065561) and three down-regulated lncRNAs (XR_600374, XR_590196, and XR_597099) were verified by quantitative reverse-transcription polymerase chain reaction. This provides new insight into the potential role of lncRNAs in the pathogenesis of OSA-induced cardiovascular disease [75] (Table 2).

## 5. DNA Methylation

Methylation of cytosine occurs at the 5th position of the pyrimidine ring in the sequence context of cytosine followed by guanine (CpG), and is an inheritable and reversible epigenetic change, which silences gene transcription by altering accessibility for transcription factors and RNA polymerases in the gene promoter region, and activates gene transcription by alternative splicing in the gene body region [77]. In mammalian somatic cells, around 75% of CpG dinucleotides are methylated, while a specific category of CpG-rich sequences termed CpG islands are generally unmethylated within the promoter region and moderately methylated within the enhancer region [78].

### 5.1. Aberrant DNA Methylation Patterns in OSA Patients 

The *FOXP3* gene, which regulates expression of T regulatory lymphocytes, displayed increased DNA methylation among children with OSA who exhibit increased systemic inflammatory responses, suggesting that epigenetically mediated down-regulation of specific T regulatory lymphocytes may be an important determinant of the inflammatory and morbidity phenotype in OSA [79]. The presence of abnormal eNOS-dependent vascular responses in children with OSA has been shown to be associated with DNA hypermethylation in the *eNOS* gene promoter region, suggesting that an epigenetic mechanism may be responsible for endothelial dysfunction in pediatric patients with OSA. [80]. In our previous epigenome-wide DNA methylation analyses, we identified and validated several novel differentially methylated genes associated with OSA and its adverse consequences. We found that Interleukin1 receptor 2 (*IL-1 R2*) hypomethylation and Androgen receptor (*AR*) hypermethylation may constitute an important determinant of disease severity, while Natriuretic peptide receptor2 (*NPR2*) hypomethylation and speckled protein 140 (*SP140*) hypermethylation may provide a biomarker for vulnerability to excessive daytime sleepiness in OSA [81]. Furthermore, four days of IHR treatment in vitro (seven h of alternative 0% and 21% O2 each day) did not result in significant alteration of these DNA methylation levels. Previous studies have shown that increased IL-1R2 expression indicates the activation of endogenous pathways of negative regulation of inflammation, while inhibition of androgen action via the 5 α-reductase pathway may alleviate breathing instability during sleep. Since existing data suggest that, overall, the methylome is stable beyond adolescence [78], we speculate that DNA methylation changes occurring in the prenatal period or early life may predispose subjects to an epigenotype with hypomethylation of the *IL-1R2* gene promoter and hypermethylation of the *AR* gene promoter, and subsequently a phenotype with more frequent hypoxic events during sleep in adulthood. Alterations induced by hyperammonemia on NPR2-cGMP pathways in the brain have been reported to result in hepatic encephalopathy characterized by altered sleep-wake patterns and drowsiness, while SP140 has been shown to play a role in pro-inflammatory interferon response. Therefore, we speculate that hypomethylation over the six gene promoter regions involved in the *NPR2* and *SP140* pathways may play a crucial role in the development of the excessive daytime sleepiness phenotype in OSA. However, it is also possible that chronic IHR triggered these DNA methylation changes, which in turn lead to frequent hypoxia and sleepiness phenotypes in OSA. (Table 3)

### 5.2. Aberrant DNA Methylation Patterns in Animals Exposed to CIH 

CIH-exposed neonatal rats exhibited increased DNA methylation at the superoxide dismutase (*Sod 2*) gene promoter region, a modification that had lasting impact on exaggerated chemoreflexes, irregular breathing with apneas, and hypertension in adults [82]. Furthermore, the persistent adverse effects of CIH on blood pressure, breathing and carotid body chemosensory reflex were partly the result of a long-lasting suppression of anti-oxidant enzyme (AOE) genes by DNA methylation, including *Sod1*, *Sod2*, thioredoxin reductase (*Txnrd2*), and peroxiredoxin 4 (*Prdx4*) [49]. Endothelial cells isolated from mesenteric arteries of CIH-exposed neonatal mice expressed lower DNA methylation levels of angiotensin-converting enzyme (*ACE*) and angiotensinogen (*ATG*) gene promoter regions in association with higher systolic blood pressure, impaired baroreflex responses, and decreased heart rate variability at adulthood [83]. The results indicate that neonatal exposure to IH may predispose the animals to autonomic dysfunction at adulthood through enhanced ROS production caused by hypermethylation over the AOE gene promoter regions, and through diminished vasodilatory responses caused by hypomethylation of the *ACE*/*ATG* genes, while CIH may lead to similar DNA methylation changes of the AOE genes (Table 3).

Additionally, exposure to CIH in mice engrafted with epithelial lung cancer cells increased tumor size, weight and invasiveness, as well as the shedding of DNA into circulation, which carried epigenetic modifications, such as hypermethylation of the tumor-promoting *Rab3a* gene [84] (Table 3).

## 6. Crosstalk between microRNA and DNA Methylation/Histone Modification under Sustained Hypoxia Stimuli

Posttranslational histone modifications and DNA methylation act together to regulate gene expression by controlling the chromatin state through affecting accessibility of DNA for transcription factors/other regulators. Emerging evidence suggest that crosstalk between microRNA and DNA methylation/histone modification could offer potential biomarkers and targeted therapies in OSA [85].

MiRNA genes can be regulated through epigenetic silencing with histone modification and/or DNA methylation of CpG islands. To date, 180 oncogenic miRNA genes have been proven to be regulated by DNA methylation in 36 different cancer types [86], and some hypoxia-related miRNA genes are regulated by aberrant DNA methylation. DNA demethylation regulates the expression of miR-210 in neural progenitor cells subjected to hypoxia [87], while hypoxia-induced miR-210 promoter demethylation enhances proliferation, autophagy, and angiogenesis of schwannoma cells. miR-34a gene promoter hypomethylation was revealed in all preeclamptic placentas, and became hypermethylated under hypoxic stimuli [88]. DNA methylation of the CpG islands in the promoter region of miR-124-3p was increased under hypoxic conditions [89]. At present, there is no study dealing with the role of aberrant DNA methylation of miRNA genes in IHR-mediated injury. The investigation of the role of aberrant DNA methylation and histone modifications of specific miRNAs in OSA is still recent and rather limited, requiring further studies.

## 7. Limitations and Perspectives of the OSA Cohort Studies, in vitro IHR Experiments, and CIH Animal Models

### 7.1. Limitations of Past Epigenetic Studies on OSA

A cause and effect relationship between aberrant DNA methylation and OSA cannot be determined directly based on the results of cohort studies, because inherited DNA methylation patterns (epigenotype) may affect the development of OSA, and environmental stimuli may cause OSA progression through DNA methylation changes. The reversibility with treatment and the results of in vitro IHR experiments may help clarify this relationship, although a lot of confounding factors can affect changes in epigenetic markers and need to be adjusted. One limitation of the in vitro experiments investigating IHR is the lack of consensus and homogeneity in the magnitude, frequency, and duration of IHR exposures, because an ideal in vitro IHR model mimicking chronic IHR in OSA is still lacking. Second is the lack of consideration of any additive effect of IHR plus concomitant intermittent hypercapnia, which is also an essential hallmark of severe OSA events. On the other hand, animal studies have demonstrated that intermittent hypoxia during neonatal life predisposed to enhanced hypoxic sensing in the carotid body and more sleep apnea events in adulthood, which involved DNA hypermethylation of the *Sod2* gene promoter region resulting in high loop gain of the ventilatory control system. It remains to be elucidated whether this epigenotype applies to OSA patients. No epigenotype has been linked to the other three physiological pathways leading to OSA, including high collapsibility of upper air way, low pharyngeal negative pressure for triggering arousal from sleep, and low reflex responses of upper airway dilator muscles [90]. Animal models are essential to biological research but not predictors of human response, because of the intrinsic differences between humans and other species [91]. Besides, an ideal animal model mimicking all the pathophysiological changes in OSA patients, including IHR, intermittent hypercapnia, alternating sympathetic/parasympathetic hyperactivity, and sleep fragmentation during sleep apnea and arousal events at a frequency of more than 20 events/hour, is still lacking. No candidate miRNA of OSA has been approved by all the three investigations, namely cohort studies, in vitro IHR experiments, and CIH animal models, while candidate histone modifications coupling with specific modifying enzyme changes have not yet been identified in OSA or IHR model.

### 7.2. Perspectives of Future Epigenetic-Related Investigations in OSA

Whole-genome 5-methylcytosine profiling has uncovered dynamic hypermethylation at enhancers/promoters and dynamic global hypomethylation over extended transcriptionally inactive partially methylated domains in aged and diseased tissues, indicating essential regulatory roles for DNA methylation in the development of human diseases, including OSA [92]. In the future, single-cell multi-omic strategies will enhance the specificity and sensitivity of the analysis of DNA methylation patterns over both CpG and non-CpG sites, while site-specific methylome editing will serve as a key technique for the study of 5-methylcytosine function [78]. Histone modifications have been shown to play critical roles in nearly all biological processes including DNA repair, gene transcription and translation, and many human diseases, such as cancer, autoimmune disease, and neurodegenerative disease [93]. The link between histone modifications and OSA is beginning to be discovered. Because histone modifications are highly druggable targets, they represent an attractive target for pharmaceutical intervention. Recent studies have shed light on the complexity of miRNA-mediated gene regulation, with many factors contributing to the activity of miRNAs, including sub-cellular location, miRNA/messenger RNA abundance, miRNA over miRNA response elements affinity, cell type/state, and the availability of various miRNA-induced silencing complex components, which need to be taken into account in the design of in vitro IHR experiments [94]. In the past few years, therapeutic miRNA and small interfering RNA (siRNA) are considered one of the most significant therapeutic breakthroughs in pharmaceutical development, with the highest number of patents being granted for cancer [95]. When clinical associations and biologic functions of the candidate miRNA are verified consistently, it may be the next step to initiate clinical trials using miRNA/siRNA-based therapeutics through efficient in vivo delivery systems to target specific cells and tissues under chronic IHR in OSA patients.

## 8. Conclusions

In this review, we have highlighted the growing importance of the epigenetic regulation of adaptive and maladaptive responses to IHR in OSA. Figure 1 shows proposed model of the epigenetic regulations in OSA based on the results of the cohort studies, in vitro IHR experiments, and CIH-exposed animal studies. We need to understand this heterogeneity and mechanisms that underlie systemic responses to IHR, intermittent hypercapnia, sleep fragmentation, and alternating sympathetic/parasympathetic hyperactivity in short and long time domains. In this regard, of particular interest is the potential for epigenetic mechanisms to regulate oxidative stress, cardiovascular remodeling, cognitive dysfunction, and inflammation responses acutely and throughout life in OSA. Epigenetic regulation of various end organs during chronic IHR is an important area for future research.

## Figures and Tables

**Figure 1 ijms-20-02937-f001:**
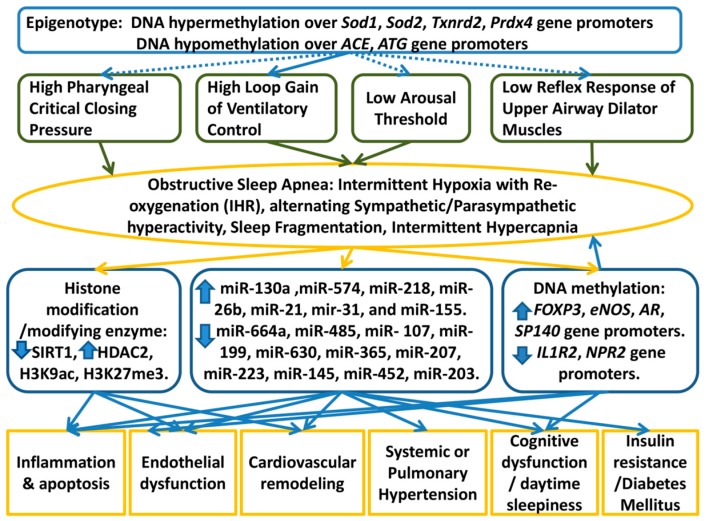
Proposed model of the roles of epigenetics-mediated regulations in the development of OSA and its clinical phenotypes. Continuous lines and arrows represent proposed cause and effect relationships based on the findings from the cohort studies, in vitro experiments, or animal models, while dotted lines and arrows represent hypothetical relationships not approved by any investigation.

**Table 1 ijms-20-02937-t001:** Selected histone modifications/modifying enzymes and their roles in pathology and clinical phenotypes of OSA.

Histone Modification or Modifying Enzyme	Attribute	Up- or Down-Regulation in Response to IHR	Investigation Models	Main Role of the Histone Modification or Modifying Enzyme	Reference
SIRT1	Class III HDAC	Down	Peripheral blood mononuclear cells of OSA patients	Repress endothelial nitric oxide synthase; negatively correlated with AHI and minimum SaO2	[55,56]
HDAC2	Class I HDAC	Up	Visceral fat of OSA patients		[57]
H3K9Ac	Active histone mark	Increased enrichment over *Foxo4*, *Lrtm2*, *Gabbr2*, *ABca1*, *IL6*	Aortic macrophages of rats, CIH for 20 weeks	Activate pro-inflammatory genes	[58]
H3K27me3	Repressive histone mark	Increased enrichment over *PPAR*/*RXR* and *LXR*/*RXR*	Aortic macrophages of rats, CIH for 20 weeks	Repress anti-inflammatory genes	[58]

**Table 2 ijms-20-02937-t002:** Selected miRNAs and Their Roles in Pathology and Clinical Phenotypes of OSA.

	Up- or Down-Regulation in Response to IHR	Investigation Models	Main Role of the miR	TARGET GENE *	Reference
miR-378a-3p	up	Plasma of OSA patients with resistant hypertension	Predict blood pressure decreases to CPAP treatment		[60]
miR-100-5p	up	Plasma of OSA patients with resistant hypertension	Predict blood pressure decreases to CPAP treatment		[60]
miR-486-5p	down	Plasma of OSA patients with resistant hypertension	Predict blood pressure decreases to CPAP treatment		[60]
miR-664a	down	Serum of OSA patients	A marker of atherosclerosis; negatively correlated with AHI		[61]
miR-130a	up	Blood of OSA patients; human umbilical vein endothelial cell	Potentiate pulmonary hypertension	*GAX*	[62]
miR-485-5p	down	Serum of OSA patients			[63]
miR-107	down	Serum of OSA patients			[63]
miR-199-3p	down	Serum of OSA patients			[63]
miR-574-5p	up	Serum of OSA patients			[63]
miR-630	down	Pediatric OSA patients; human microvascular endothelial cells	Attenuate endothelial dysfunction	*Nrf2*, *AMP kinase*, and tight junction pathways	[27]
miR-223	down	Rats, CIH for 6 weeks	Attenuate pulmonary hypertension		[68]
miR-21	up	Rats, CIH for 30 days	Induce atrial remodeling and fibrosis	*Spry1*/*ERK*/*MMP-9*; *PTEN*/*PI3K*/*AKT*	[69,70]
miR-155	up	Mice, CIH for >4 weeks; in vitro HK-2 cells, IHR	Promote kidney injury	*FOXO3a*	[72]
miR-31	up	In vitro H9c2 cardiomyocyte, IHR	Promote cardiac hypertrophy	*PKC*ε	[71]
miR-145	down	Canines, CIH for 12 weeks	Attenuate aortic remodeling and sympathetic nerve sprouting	*Smad3*	[76]
miR-365	down	In vitro Hepatocyte/macrophage, IHR	Inhibit inflammation	*IL-6*	[29]
miR-218	up	In vitro mice aortic endothelium, IHR	Potentiate apoptosis	*Robo1*	[65]
miR-26b	up	Rats, CIH for 4 weeks	Promote cognitive dysfunction		[66]
miR-207	down	Rats, CIH for 4 weeks	Attenuate cognitive dysfunction		[66]
miR-452	down	mouse 3T3-L1 and human SW872 adipocytes, IHR	Attenuate insulin resistance	*RETN*, *TNF-α*, and *CCL2*	[73]
miR-203	down	Human JHH5, JHH7, and HepG2 hepatocytes, IHR	Potentiate insulin resistance	*SELENOP HIP/PAP*	[74]

* specific messenger RNA targets regulated by the candidate miRNA.

**Table 3 ijms-20-02937-t003:** Selected differentially methylated loci and their roles in pathology and clinical phenotypes of OSA.

Genes	Hyper- or Hypo-Methylation in OSA or in Response to IHR	Investigation Model	Main Role of the Aberrant DNA Methylation	References
*FOXP3*	Hypermethylated intron1 region (mean of 11 CpG sites)	Pediatric OSA patients with high hypersensitivity CRP	Positively correlated with AHI and hypersensitivity CRP	[79]
*eNOS*	Hypermethylated promoter region (-171 CpG site)	Pediatric OSA patients with endothelial dysfunction	Decreased *eNOS* gene expression	[80]
*IL1R2*	Hypomethylated promoter region (-114 CpG site)	Adult OSA patients	Increased IL1R2 protein expression; negatively correlated with oxygen desaturation index	[81]
*AR*	Hypermethylated promoter region (-531 CpG site)	Adult OSA patients	Positively correlated with AHI	[81]
*NPR2*	Hypomethylated promoter region (-608/-618 CpG sites)	Adult OSA patients with excessive daytime sleepiness	Increased NPR2 and CNP protein expressions; negatively correlated with Epworth Sleepiness Scale	[81]
*SP140*	Hypermethylated promoter region (-194 CpG site)	Adult OSA patients with excessive daytime sleepiness	Decreased SP140 protein expression; positively correlated with Epworth Sleepiness Scale	[81]
Anti-oxidant enzymes (AOE genes: *Sod1*, *Sod2*, *Txnrd2*, *Prdx4*)	Hypermethylated promoter regions	Rats, CIH for 30 days; neonatal rats, IH from postnatal day 1 to day 10	Inhibit AOE genes, increase ROS production, exaggerate chemoreflexes of the carotid body	[49,82]
*Rab3a*	Hypermethylated promoter regions	Mice engrafted with TC1 epithelial lung cancer cells, CIH for 2 weeks followed by tumor engraftment for 4 weeks	Increased tumor growth and invasion	[84]
*Ace1*	Hypomethylated promoter regions	mesenteric endothelial cells of mice, CIH for the first 4 weeks of life	Increased *Ace* gene expression; diminished vasodilatory responses, increased ROS content	[83]
*Atg*	Hypomethylated enhancer regions	mesenteric endothelial cells of mice, CIH for the first 4 weeks of life	Increased angiotensinogen (Atg) protein expression; diminished vasodilatory responses, increased ROS content	[83]

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
