# Peer review of "Epigenetics: A Potential Mechanism Involved in the Pathogenesis of Various Adverse Consequences of Obstructive Sleep Apnea"

_ijms, 2019, doi:10.3390/ijms20122937_

Reviewer 1 Report

Although the review concerning epigenetics in sleep apnea syndrome seems very interesting, a number of points need clarifying and certain statements require further justification. These are given below.

1.       The authors should carefully read “Instructions for Authors” and write according to the instructions. For example, Title and sub-title, “Epigenetic mechanisms underlying IHR injury in OSA” should be described, “Epigenetic Mechanisms underlying IHR Injury in OSA”.

2.       In line 75, why “phenotype” and “DNA sequence” were underlined?

3.       In Table 2, [58] was omitted in miR-452. 

4.       In Table 2, “miR-203” (Uchiyama, T, et al.Biochem. Bihys. Rep.11,130-137, 2017) should be added. 

5.       Reference style should be changed according to “Instructions for Authors”.

6.       Figure 1 should be much more improved. The figure should provide a perspective view to the readers.

Author Response

Although the review concerning epigenetics in sleep apnea syndrome seems very interesting, a number of points need clarifying and certain statements require further justification. These are given below.

Ans.:  We are delighted for the opportunity to respond to the comments of the reviewer 1 on our work. Our revised manuscript has been improved by addressing these comments, and our point-by-point responses to the reviewer critiques are listed below:

1.                   The authors should carefully read “Instructions for Authors” and write according to the instructions. For example, Title and sub-title, “Epigenetic mechanisms underlying IHR injury in OSA” should be described, “Epigenetic Mechanisms underlying IHR Injury in OSA”.

Ans.: Thank you very much for your comments. As you suggest, we change the lower case into capital letters of the several words in the title.

2.                   In line 75, why “phenotype” and “DNA sequence” were underlined?

Ans.: Thank you. The underlines are deleted now.

3.                   In Table 2, [58] was omitted in miR-452. 

Ans.: Thank you. The reference is added now.

4.                   In Table 2, “miR-203” (Uchiyama, T, et al.Biochem. Bihys. Rep.11,130-137, 2017) should be added. 

Ans.: Thank you. As suggest, we add this article into the text and references.

5.                   Reference style should be changed according to “Instructions for Authors”.

Ans.: Thank you. We prepared the references with Endnote, using the recent MDPI style copy, and correct some errors in the previous version.

6.                   Figure 1 should be much more improved. The figure should provide a perspective view to the readers.

Ans.: Thank you for your comments. We sort out all the evidences together supporting the roles of epigenetic regulations in OSA in the Figure 1, and make some changes to it according to the contents of the text.

Reviewer 2 Report

Obstructive sleep apnea(OSA) is a significant health problem associated with several important comorbidities such as cardiovascular diseases and metabolic dysfunctions. Chronic intermittent hypoxia (CIH) is a hallmark of OSA and considered a major contributor to the adverse consequences of OSA. In this manuscript, Yung-Che Chen et al attempted to provide an overview of recent advances on epigenetic mechanisms underlying pathogenesis of inflammation, oxidative stress, and sympathetic hyperactivity associated with OSA. This manuscript covers an important topic given the high prevalence of OSA and lack of optimal treatment of OSA. However, I have some concerns which need to be addressed.

 1.     The manuscript is not focused enough on CIH associated with OSA such that readers easily get confused with epigenetic regulation under CIH, sustained hypoxia and hypoxia/re-oxygenation injury.  For example, authors made a great effort to discuss epigenetic regulation during sustained hypoxia (lines 53-62; lines 135-153; lines 313-331), which does not help to understand epigenetic mechanisms underlying CIH. Likewise, Authors quoted Refs 30 and 41 to describe histone modification in OSA(lines 164-169), which is totally misleading, because the working models in the studies are hypoxia (24 hours)/re-oxygenation (1 or 2 hrs).  

2.     Authors might need to discuss more how OSA leads to oxidative stress, inflammation, and sympathetic overactivation.

3.     Authors quoted a reference (Ref.3) and stated in “Introduction” (lines 49-52) that the use of CPAP was not associated with reduced risk of cardiovascular outcomes, and diabetes mellitus, thus it is important to identify the best pharmacological approach to treat OSA. If this statement is true, what was the purpose to describe the works later (Refs. 38&39, lines 156-160; Ref. 44, lines 199-201) showing beneficial outcomes of CPAP treatment?

4.     Line 219, I don’t think Ref.33 is the correct one.

Author Response

Obstructive sleep apnea(OSA) is a significant health problem associated with several important comorbidities such as cardiovascular diseases and metabolic dysfunctions. Chronic intermittent hypoxia (CIH) is a hallmark of OSA and considered a major contributor to the adverse consequences of OSA. In this manuscript, Yung-Che Chen et al attempted to provide an overview of recent advances on epigenetic mechanisms underlying pathogenesis of inflammation, oxidative stress, and sympathetic hyperactivity associated with OSA. This manuscript covers an important topic given the high prevalence of OSA and lack of optimal treatment of OSA. However, I have some concerns which need to be addressed.

Ans.:  We are delighted for the opportunity to respond to the comments of the reviewer 2 on our work. Our revised manuscript has been improved by addressing these comments, and our point-by-point responses to the reviewer critiques are listed below:

1.            The manuscript is not focused enough on CIH associated with OSA such that readers easily get confused with epigenetic regulation under CIH, sustained hypoxia and hypoxia/re-oxygenation injury.  For example, authors made a great effort to discuss epigenetic regulation during sustained hypoxia (lines 53-62; lines 135-153; lines 313-331), which does not help to understand epigenetic mechanisms underlying CIH. Likewise, Authors quoted Refs 30 and 41 to describe histone modification in OSA(lines 164-169), which is totally misleading, because the working models in the studies are hypoxia (24 hours)/re-oxygenation (1 or 2 hrs).  

Ans.: Thank you very much for your comments. As opposed to a great deal of evidences showing the role of epigenetic regulation in sustained hypoxia-induced cell injury, only a few references related to IHR, CIH, or OSA could be found. As you suggest that the working model used by the two studies was flawed by lacking cycling hypoxia and did not mimic OSA, we delete these two references.

2.            Authors might need to discuss more how OSA leads to oxidative stress, inflammation, and sympathetic overactivation.

Ans.: Thank you for your comments. We add a statement about how OSA leads to oxidative stress, inflammation, and sympathetic overactivation in the section 2.1.

3.            Authors quoted a reference (Ref.3) and stated in “Introduction” (lines 49-52) that the use of CPAP was not associated with reduced risk of cardiovascular outcomes, and diabetes mellitus, thus it is important to identify the best pharmacological approach to treat OSA. If this statement is true, what was the purpose to describe the works later (Refs. 38&39, lines 156-160; Ref. 44, lines 199-201) showing beneficial outcomes of CPAP treatment?

Ans.: Thank you very much for your comment. Actually, randomized control trials have demonstrated that CPAP treatment improves daytime sleepiness, insulin resistance, and hypertension, but it does not reduce the risk of cardiovascular events, diabetes mellitus, and death. One of the reasons may be insufficient time of CPAP use during sleep every night, for instance, less than 4 hour/night. We make a change to the statement as the following: “CPAP clearly improves vigilance and cognitive function, reduces insulin resistance, and is the most effective in lowering blood pressure in OSA patients with refractory hypertension, but the use of CPAP was not associated with reduced risks of cardiovascular outcomes, diabetes mellitus, or death for patients with OSA in recent randomized control trials.

4.            Line 219, I don’t think Ref.33 is the correct one.

Ans.: Thank you for your comment. As suggest, we remove this reference and add a correct one.

Round  2

Reviewer 1 Report

"Mir 103/107 regulates lipid metabolism by inhibiting betatrophin/ANGPTL8" was published by Mohamed Abu-Farha et al. in ADA Meeting 2019, San Francisco, CA. The review paper will be more perfect by including the paper, I think.   

Reviewer 2 Report

I don't  have further comments.